

SciPost Phys. Lect. Notes 103 (2025)

# Stochastic resetting and large deviations

**Martin R. Evans⋆ and John C. Sunil†**

SUPA, School of Physics and Astronomy, University of Edinburgh,
Peter Guthrie Tait Road, Edinburgh, EH9 3FD, United Kingdom

⋆ m.evans@ed.ac.uk , † j.chakanal-sunil@sms.ed.ac.uk

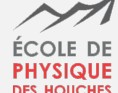

*Part of the 2024-07: Theory of Large Deviations and Applications collection*
*Session 123 of the Les Houches School, July 2024*
*published in the Les Houches Summer School Lecture Notes series*

## Abstract

**Stochastic resetting has been a subject of considerable interest within statistical physics, both as means of improving completion times of complex processes such as searches and as a paradigm for generating nonequilibrium stationary states. In these lecture notes we give a self-contained introduction to the toy model of diffusion with stochastic resetting. We also discuss large deviation properties of additive functionals of the process such as the cost of resetting. Finally, we consider the generalisation from Poissonian resetting, where the resetting process occurs with a constant rate, to non-Poissonian resetting.**



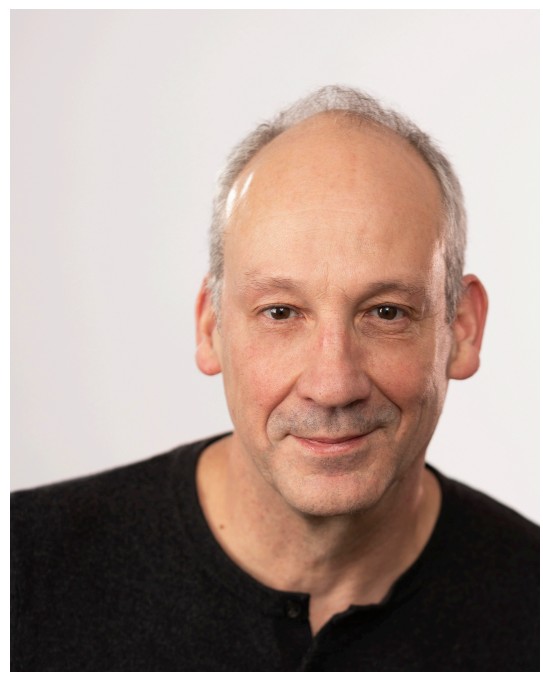

# 1 Introduction

These are the lecture notes from two lectures given at the 2024 summer school on Large Deviations held at Les Houches School of Physics. The lectures give an introduction to diffusion with stochastic resetting and how this relates to the general theme of the school, large deviations.

In recent years there has been considerable interest in stochastic resetting for a number of reasons. As we shall motivate below, resetting can speed up some complex task such as a search process by cutting off errant trajectories that go wandering off in the wrong direction. Resetting also provides a paradigm for creating a nonequilibrium state, by virtue of continually restarting the dynamics and not letting the system relax to equilibrium. Perhaps the simplest dynamical process, to which one can add a resetting process, is diffusion. Indeed, diffusion with stochastic resetting [1] has become an important toy model where an in-depth understanding of resetting can be obtained from exact calculations of relevant quantities such as survival probabilities, mean first passage times etc.

The aim of these lecture notes is to provide a self-contained introduction to diffusion with stochastic resetting that will allow the reader to execute the calculations in full, using straightforward techniques such as Laplace transforms and renewal equations. These notes will hopefully be complementary to Reference [2], which is a more detailed review and contains a comprehensive bibliography up to 2020. The present notes also contain some more recent material. We also note that a historical perspective on resetting can be found in [3] and a brief review is given in [4].

These notes are organised as follows. After motivating the study of stochastic resetting in Section 1.1, we begin by reviewing the diffusion equation in Section 2 and first passage times 3 and Laplace transform techniques. We then use these techniques to study diffusion with Poissonian resetting in Section 4 and diffusion with resetting and an absorbing target in Section 5. We discuss large deviations of additive functionals of diffusion with resetting in Section 6 and the the example of the cost of resetting in Section 7. Finally we generalise to non-Poissonian resetting, defined by a waiting time distribution between resets, in Section 8.

## 1.1 Motivations

The process of stochastic resetting can be motivated by a simple everyday experience of searching for misplaced keys. During the search process, we periodically reset back to the place where the keys usually should be and restart the search. It turns out that such resetting expedites the completion time of the search process [1, 2]. More generally, motivations for studying stochastic resetting are

(i) **Searching for a target:** In search processes it has been established that mixing local moves with long range moves improves the search process. This is termed as an "Intermittent search strategy" [5]. In the case of searches with stochastic resetting, the local moves are diffusion and the long range moves are the resets.

(ii) **Expediting completion times:** More generally, it has been observed that restarting a complex task, which has a long time tail of completion times, can speed up performance. For example, consider a complicated chemical reaction [6], such as,

$$\text{E} + \text{S} \underset{k_{\text{off}}}{\overset{k_{\text{on}}}{\rightleftharpoons}} \text{ES} \xrightarrow{\tau} \text{P}.$$

The above equation represents a reaction where $E$ is the enzyme, and $S$ is the substrate which combines to form $ES$ through a binding rate $k_{\text{on}}$ and then takes a random time $\tau$ (which could have long tails in its distribution) to produce the product $P$. The rate of

unbinding of the enzyme, $k_{\text{off}}$, can be thought of as the resetting rate, and it turns out that having such a rate enhances the completion time of the process by cutting off the long tails in the distribution of $\tau$.

(iii) **Generating non-equilibrium steady states (NESS):** Resetting the system to its initial state prevents the system from relaxing to its equilibrium state. The resetting process generates a probability current back to the initial condition from all other configurations, thus ensuring detailed balance is not satisfied. This results in a simple, but non-trivial, NESS exhibiting circulation of probability.

# 2 Preliminaries: Diffusion equation and solution

Diffusion is perhaps the simplest dynamical process in nature. Our aim is to study how introducing resetting fundamentally changes the behaviour of a diffusive particle.

## 2.1 Diffusion equation

We begin by deriving the diffusion equation. Consider an over-damped diffusing particle starting at $x(t = 0) = x_0$, which can be represented as,

$$x_{t+\Delta t} = x_t + \xi, \tag{1}$$

where $\xi$ is a Gaussian white noise given by the distribution

$$P(\xi) = \frac{1}{\sqrt{4\pi D \Delta t}} e^{-\frac{\xi^2}{4D\Delta t}}. \tag{2}$$

Averaging over all possible events between $t$ and $t + \Delta t$, we write

$$P(x, t + \Delta t | x_0) = \langle P(x - \xi, t | x_0) \rangle, \tag{3}$$

where the angle brackets indicates an average over the noise $\xi$. Expanding for small $\xi$, we obtain

$$P(x, t + \Delta t) = \left[ P(x, t) - \frac{\partial P(x, t)}{\partial x} \langle \xi \rangle + \frac{1}{2} \frac{\partial^2 P(x, t)}{\partial x^2} \langle \xi^2 \rangle + \ldots \right] \tag{4}$$

$$= P(x, t) + \Delta t \left[ D \frac{\partial^2}{\partial x^2} P(x, t) + \ldots \right], \tag{5}$$

where in (4),(5) we have suppressed the dependence on $x_0$ for ease of presentation. In going to equation (5), we have made use of the following properties of white noise

$$\langle \xi(t) \rangle = 0,$$
$$\langle \xi(t)\xi(t') \rangle = 2D\Delta t \, \delta(t - t').$$

Dividing (5) by $\Delta t$ and taking $\Delta t \to 0$, yields the diffusion equation

$$\frac{\partial}{\partial t} P(x, t | x_0) = D \frac{\partial^2}{\partial x^2} P(x, t | x_0). \tag{6}$$

In a similar way we can average over events between $t = 0$ and $t = \Delta t$ and write

$$P(x, t + \Delta t | x_0) = \langle P(x, t | x_0 + \xi) \rangle, \tag{7}$$

which leads to a backward equation. For diffusion this looks the same as the forward equation, where the spatial variable is now $x_0$

$$\frac{\partial}{\partial t}P(x,t|x_0) = D\frac{\partial^2}{\partial x_0^2}P(x,t|x_0). \tag{8}$$

Taking the initial condition

$$P(x,t=0|x_0) = \delta(x-x_0),$$

the solution of (6) is the familiar Gaussian

$$P(x,t|x_0) = \frac{1}{\sqrt{4\pi Dt}}e^{-\frac{(x-x_0)^2}{4Dt}}. \tag{9}$$

Perhaps the simplest method to obtain the solution (9) is to assume a scaling solution

$$P = (Dt)^{-1/2}f(z), \quad \text{where } z = \frac{(x-x_0)}{(Dt)^{1/2}}, \tag{10}$$

so that $z$ is the dimensionless scaling variable and $f(z)$ is the scaling function. The prefactor $(Dt)^{-1/2}$ in $P$ ensures normalisation of probability through

$$\int_{-\infty}^{\infty}\mathrm{d}x\,P(x,t|x_0) = \int_{-\infty}^{\infty}\mathrm{d}z\,f(z) = 1. \tag{11}$$

Substituting expression (10) in (6) leads to

$$-\frac{1}{2}\frac{\mathrm{d}}{\mathrm{d}z}[zf(z)] = \frac{\mathrm{d}^2}{\mathrm{d}z^2}f(z). \tag{12}$$

The solution which tends to zero as $z \to \infty$ and is normalised is $f(z) = (4\pi)^{-1/2}e^{-z^2/2}$.

## 2.2 Solution by Laplace transform

Let us also derive the solution of the diffusion equation by Laplace transform and discuss the subsequent inversion using a Bromwich contour to obtain the diffusion propagator. The reasons for choosing this method will become apparent when we move on to more complicated problems. We define the Laplace transform with respect to time for the PDF as,

$$\widetilde{P}(x,s|x_0) = \int_0^{\infty}dt\,e^{-st}P(x,t|x_0). \tag{13}$$

By making use of integration by parts, we can rewrite the Laplace transform of the time derivative of $P(x,t|x_0)$ as

$$\int_0^{\infty}e^{-st}\frac{\partial P}{\partial t}dt = [e^{-st}P]_0^{\infty} + s\int_0^{\infty}e^{-st}Pdt$$
$$= -\delta(x-x_0) + s\widetilde{P}. \tag{14}$$

The space derivative goes through unaltered as the Laplace transform only acts on the time variable. Hence we obtain the Laplace transform of the diffusion equation as

$$D\frac{\partial^2}{\partial x^2}\widetilde{P}(x,s|x_0) - s\widetilde{P}(x,s|x_0) = -\delta(x-x_0). \tag{15}$$

The solution of (15) is

$$\widetilde{P}(x,s|x_0) = \frac{1}{\sqrt{4sD}} e^{-\sqrt{\frac{s}{D}}|x-x_0|},\tag{16}$$

which can be verified by making use of the identity

$$\frac{d^2}{dx^2}\left[e^{-a|x-b|}\right] = -2a\delta(x-b) + a^2 e^{-a|x-b|}.\tag{17}$$

Finally, to obtain the solution back in the time domain, we have to invert the Laplace transform by making use of the Bromwich integral

$$P(x,t|x_0) = \frac{1}{2\pi i}\int_{\gamma-i\infty}^{\gamma+i\infty} ds\, e^{st}\widetilde{P}(x,s|x_0),\tag{18}$$

where $\gamma$ is a real number which lies to the right of all the singularities of the function to be inverted in the complex $s$ plane. The singularity in (16) is a branch point at $s = 0$ due to the presence of $s^{1/2}$. Therefore the inversion integral for (16) has to be evaluated around a branch cut along the negative real axis, which will finally result in (9) (See Appendix A for details). Rather than performing the contour integral, a simpler method to obtain the final result is to make use of a known integral (a particular case of identity 3.174.9 from [7], see Appendix B for an elementary proof)

$$\int_0^\infty dt\, t^{-1/2} e^{-\frac{\beta}{t}-st} = \frac{\pi^{1/2}}{s^{1/2}} e^{-2(\beta s)^{1/2}}.\tag{19}$$

Then we compare (16) and (19) and match $\beta = \frac{(x-x_0)^2}{4D}$. One then deduces that the function whose Laplace transform is (16) is (9).

## 3 Preliminaries: Diffusion with an absorbing target

Now consider the situation where in addition to a diffusing particle, there is an absorbing target [2, 8] at $x = x_T$. When the particle touches the target it is absorbed and the process is considered to be completed. We will usually take the absorbing target to be at the origin so that $x_T = 0$.

### 3.1 Survival probability

We are interested in the probability that the particle survives up to a time $t$. Here we define $P(x,t|x_0)$ as the joint probability density for the particle position $x$ at time $t$ having started from $x_0$, and the particle not having been absorbed. The presence of the absorbing target at $x_T$ can be modelled as a boundary condition $P(x_T,t|x_0) = P(x,t|x_T) = 0$.

The backward Fokker-Planck equation reads

$$\frac{\partial}{\partial t}P(x,t|x_0) = D\frac{\partial^2}{\partial x_0^2}P(x,t|x_0).\tag{20}$$

The survival probability $q(t|x_0)$ is obtained by integrating over all final positions at time $t$

$$q(t|x_0) = \int_0^\infty dx\, P(x,t|x_0).\tag{21}$$

Integrating over $x$ from 0 to $\infty$, and exchanging the order of differentiation and integration (since integration is with respect to $x$ while the derivatives are with respect to $x_0$), we end up with an equation for the survival probability as

$$\frac{\partial}{\partial t} q(t|x_0) = D \frac{\partial^2}{\partial x_0^2} q(t|x_0) \,. \tag{22}$$

Taking the target to be at the origin, $x_T = 0$, the boundary condition becomes

$$q(t|0) = 0 \,, \tag{23}$$

and initial condition becomes

$$q(0|x_0) = 1 \,. \tag{24}$$

The solution to (22) with the boundary and initial conditions (23, 24) is given by

$$q(t|x_0) = \mathrm{erf}\left(\frac{x_0}{\sqrt{4Dt}}\right) \,, \tag{25}$$

where the error function is defined as

$$\mathrm{erf}(z) = \frac{2}{\sqrt{\pi}} \int_0^z du \, e^{-u^2} \,. \tag{26}$$

The simplest way to obtain (25) is to assume a scaling form $q(t|x_0) = g(z)$ where the scaling variable is now $z = x_0/\sqrt{4Dt}$. Substituting in (22) one obtains

$$-2z \frac{\mathrm{d}g}{\mathrm{d}z} = \frac{\mathrm{d}^2 g}{\mathrm{d}z^2} \,. \tag{27}$$

Integrating twice and using the boundary conditions $g(0) = 0$, $g(\infty) = 1$ yields (25).

Again, one can also use the method of Laplace transform to obtain this solution. The Laplace transform of (22) is given by

$$D \frac{\partial^2 \widetilde{q}}{\partial x_0^2} - s\widetilde{q} = -1 \,, \tag{28}$$

with boundary conditions $\widetilde{q}(s|0) = 0$ and $\widetilde{q}(s|\infty) = 1/s$, which can be solved to obtain

$$\widetilde{q}(s|x_0) = \frac{1 - e^{-x_0\sqrt{\frac{s}{D}}}}{s} \,. \tag{29}$$

Again, (29) can be inverted using the Bromwich integral by integrating around the branch cut at $s = 0$ to obtain (25), but is easier to use the known integral

$$\int_0^\infty \mathrm{d}t \, e^{-st} \, \mathrm{erf}\left(\frac{\beta}{t^{1/2}}\right) = \frac{1 - e^{-2\beta s^{1/2}}}{s} \,, \tag{30}$$

which can be derived using (19) (see Appendix C).

The great advantage of using the Laplace transform is that we can obtain the large $t$ behaviour quite easily, without having to invert the full Laplace transform. For this we expand $\widetilde{q}(s|x_0)$ for small values of $s$ and invert it term by term using the inverse Laplace transform identity

$$\mathcal{L}_{s \to t}^{-1}\left\{\frac{1}{s^\alpha}\right\} = \frac{1}{\Gamma(\alpha)} t^{\alpha-1} \,, \tag{31}$$

where the r.h.s. is zero when $\alpha = 0, -1, -2, \ldots$ Following this approach we get, expanding (29)

$$\widetilde{q}(s|x_0) \simeq \frac{x_0}{\sqrt{sD}} - \frac{x_0^2}{2D} + \mathcal{O}(\sqrt{s}). \tag{32}$$

The large $t$ behaviour is found using (31)

$$q(t|x_0) \simeq \frac{x_0}{(\pi D t)^{1/2}} + \mathcal{O}(t^{-3/2}), \tag{33}$$

where we have used $\Gamma(1/2) = \pi^{1/2}$.

## 3.2 Mean first passage time (MFPT)

The mean first passage time to the absorbing target is obtained from the survival probability as follows

$$\langle T(x_0) \rangle = \int_0^\infty dt \, t F(t, x_0), \tag{34}$$

where

$$F(t, x_0) = -\frac{\partial q(t|x_0)}{\partial t} \tag{35}$$

is the first passage time distribution i.e. minus the rate of change of the survival probability is the rate at which the particle is absorbed. But due to the $t^{-1/2}$ tails for the survival probability for diffusive processes as obtained in (33), $F(t, x_0) \sim t^{-3/2}$ and the integral (34) for the mean first passage time diverges. That is, the mean first passage time for a diffusive process is infinite even though the particle is absorbed with probability one!

**Note:** When the survival probability goes to zero faster than $t^{-1}$, the mean first passage time (34) can be simplified using integration by parts as

$$\begin{aligned} \langle T(x_0) \rangle &= -[q(t|x_0)t]_0^\infty + \int_0^\infty dt \, q(t|x_0) \\ &= \int_0^\infty dt \, q(t|x_0) \\ &= \tilde{q}(s = 0|x_0). \end{aligned} \tag{36}$$

We will use this expression for $\langle T(x_0) \rangle$ later on.

# 4 Diffusion with stochastic resetting

We now consider adding Poissonian resetting to the diffusive process discussed in the previous sections [1,2]. Poissonian resetting is defined by a constant resetting rate (per unit time) $r$ to a resetting site $x_r$. This leads to the following process:

$$x_{t+\Delta t} = \begin{cases} x_t + \xi, & \text{with prob. } 1 - r\Delta t, \\ x_r, & \text{with prob. } r\Delta t. \end{cases} \tag{37}$$

As for diffusion, we first write the forward master equation for the process with resetting. We denote $P_r(x, t)$ as the probability density at time $t$ under resetting at rate $r$, where we have

suppressed the dependence on initial condition $x_0$. Upon averaging over all possible events in between the time $t$ to $t + \Delta t$, we get

$$
\begin{aligned}
P_r(x, t + \Delta t) &= r\Delta t \delta(x - x_r) + (1 - r\Delta t)\langle P_r(x - \xi, t)\rangle \\
&= r\Delta t \delta(x - x_r) + (1 - r\Delta t) \\
&\quad \times \left[ P_r(x, t) - \frac{\partial P_r(x, t)}{\partial x}\langle \xi \rangle + \frac{1}{2}\frac{\partial^2 P_r(x, t)}{\partial x^2}\langle \xi^2 \rangle + \dots \right] \\
&= P_r(x, t) + \Delta t \left[ r\delta(x - x_r) - rP_r(x, t) + D\frac{\partial^2}{\partial x^2}P_r(x, t) \right] + \dots,
\end{aligned}
\tag{38}
$$

where as before the angle brackets $\langle \cdot \rangle$ indicates an average of the the noise $\xi$.

Then in the limit $\Delta t \to 0$, we obtain the equation for the evolution of $P_r(x, t)$ as

$$
\frac{\partial}{\partial t}P_r(x, t) = D\frac{\partial^2}{\partial x^2}P_r(x, t) - rP_r(x, t) + r\delta(x - x_r).
\tag{39}
$$

We thus obtain a diffusion equation with two additional terms proportional to $r$ on the r.h.s.. The first represents the loss of probability with rate $r$ from any position $x$, the second represents the gain of probability at resetting position $x_r$ with rate $r$ from all other positions.

As $t \to \infty$ we reach a stationary state $P_r^*(x)$ with $\frac{\partial P_r^*}{\partial t} = 0$. The forward equation (39) becomes

$$
D\frac{\partial^2}{\partial x^2}P_r^*(x, t) - rP_r^*(x, t) = -r\delta(x - x_r),
\tag{40}
$$

which is the same as equation (15) with $s$ replaced by $r$ and a factor of $r$ in the r.h.s.. Thus the solution is

$$
P_r^*(x) = \frac{\alpha_0}{2}e^{-\alpha_0|x - x_r|},
\tag{41}
$$

where

$$
\alpha_0 = \sqrt{\frac{r}{D}}.
\tag{42}
$$

Equation (41) is known as a one-dimensional Laplace distribution, and is illustrated in Fig. 1, where one sees a cusp at the resetting site. Due to the existence of a non-zero probability current in the system with resetting, the resulting steady state is a Non-Equilibrium Steady State (NESS).

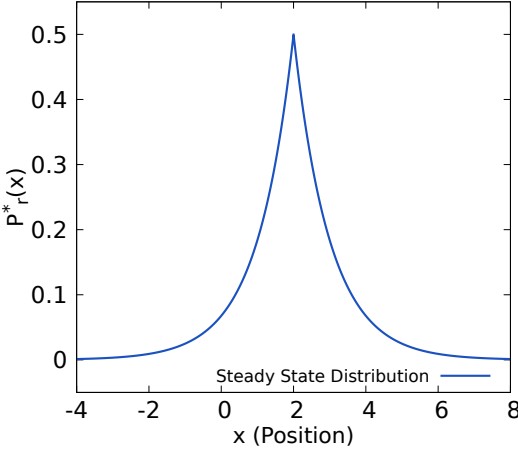

Figure 1: Steady state distribution for $\alpha_0 = 1$ and $x_r = 2$. The steady state is a Laplace distribution with a cusp at the resetting position $x_r = 2$. The distribution is symmetric about the resetting position.

Similarly, to obtain the backward master equation, we consider the evolution of the particle in the first time step from 0 to $\Delta t$. We then obtain (restoring the dependence on the initial condition),

$$P_r(x, t + \Delta t | x_0) = r\Delta t P_r(x, t | x_r) + (1 - r\Delta t)\langle P_r(x, t | x_0 + \xi)\rangle, \tag{43}$$

which upon expanding and taking the $\Delta t \to 0$ limit leads to

$$\frac{\partial}{\partial t} P_r(x, t | x_0) = D \frac{\partial^2}{\partial x_0^2} P_r(x, t | x_0) + r P_r(x, t | x_r) - r P_r(x, t | x_0). \tag{44}$$

## 4.1 Renewal equation approach

Another equivalent, but simpler, method to obtain the propagator for the system with resetting is to use the renewal approach [1,2,9]. This approach is based on the fact that between resets, the process behaves exactly like a normal diffusion process with the resetting position as the initial condition (See Fig:2).

To show this more concretely, the last renewal equation can be written as:

$$P_r(x, t | x_0) = e^{-rt} P_0(x, t | x_0) + \int_0^t dt' \, r e^{-r(t-t')} \int_{-\infty}^{\infty} dx' P_r(x', t' | x_0) P_0(x, t - t' | x_r). \tag{45}$$

The first term is the product of the probability of no resets in time $t$, $e^{-rt}$, and reaching $x$ at time $t$ without resetting, $P_0(x, t | x_0)$. The subscript 0 just denotes usual diffusion. The second term represents the probability of reaching $x$ at time $t$ when there has been at least one reset. We integrate over the time of the *last* reset, $t'$, and $dt' \, r e^{-r(t-t')}$ is the probability of a reset in time $t' \to t' + dt'$ and no further resets in time $t - t'$. We also integrate over the position $x'$ at time $t'$. However, as we have conservation of probability $\int_{-\infty}^{\infty} dx' P_r(x', t' | x_0) = 1$. Finally, changing integration variable to $u = t - t'$ the last renewal equation becomes

$$P_r(x, t | x_0) = \underbrace{e^{-rt} P_0(x, t | x_0)}_{\substack{\text{Probability of no reset} \\ \text{up to time } t \text{ and reaching } x \\ \text{by usual diffusion.}}} + \int_0^t \underbrace{du \, r e^{-ru}}_{\substack{\text{Probability of reset between} \\ t' \text{ and } t' + dt' \text{ and no resets} \\ \text{for the remaining time } t - t'.}} \underbrace{P_0(x, u | x_r)}_{\substack{\text{After the last reset,} \\ \text{the system evolves like usual} \\ \text{diffusion starting from } x_r.}}. \tag{46}$$

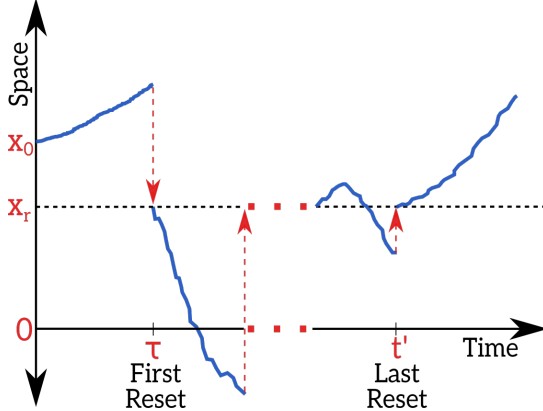

Figure 2: A schematic of trajectory for a particle undergoing stochastic resetting. The particle starts at $x_0$ and resets to $x_r$. $t'$ represents the time of last reset and $\tau$ represents the time of first reset.

To obtain the steady-state solution from the (46), we look at the $t \to \infty$ limit, which makes the first term zero, and the remaining term gives the steady state solution as

$$P_r^*(x) = r \int_0^\infty du \, e^{-ru} P_0(x, u | x_r), \tag{47}$$

which is the Laplace transform of a Gaussian, which we derived in Section 2 Eq. (16), and yields (41).

In the Laplace domain, again making use of the convolution theorem, the renewal equation (46) becomes

$$\widetilde{P}_r(x, s | x_0) = \widetilde{P}_0(x, r + s | x_0) + \frac{r}{s} \widetilde{P}_0(x, r + s | x_r), \tag{48}$$

where

$$\widetilde{P}_r(x, s | x_0) = \int_0^\infty dt \, e^{-st} P_r(x, t | x_0). \tag{49}$$

Now setting $x_0 = x_r$ so that initial position is the resetting site, we obtain

$$\widetilde{P}_r(x, s | x_r) = \frac{r + s}{s} \widetilde{P}_0(x, r + s | x_r). \tag{50}$$

This provides another route to deriving the stationary state, which is given by the final value theorem of the Laplace transform as the coefficient of $1/s$ as $s \to 0$ in (50). Thus,

$$P_r^*(x | x_r) = r \widetilde{P}_0(x, r | x_r), \tag{51}$$

which recovers (47).

## 4.2 Mean squared displacement

We can use the renewal equation (46) to compute the mean squared displacement (MSD) as a function of time. For convenience let us take $x_0 = x_r = 0$. Then multiplying (46) $x^2$ and integrating over $x$ gives an equation for the MSD under resetting, $\langle x^2(t) \rangle_r$, in terms of the MSD in the absence of resetting, $\langle x^2(t) \rangle_0 = 2Dt$:

$$\langle x^2(t) \rangle_r = e^{-rt} \langle x^2(t) \rangle_0 + \int_0^t du \, r e^{-ru} \langle x^2(u) \rangle_0 \tag{52}$$

$$= 2Dt e^{-rt} + 2Dr \int_0^t du \, u e^{-ru} \tag{53}$$

$$= \frac{2D}{r} \left( 1 - e^{-rt} \right). \tag{54}$$

Thus for $t \ll 1/r$ the particle behaves diffusively and MSD grows linearly, but for $t \gg 1/r$ the MSD saturates to its stationary value.

## 4.3 Relaxation to stationary state

The last renewal equation may also be used to study the relaxation to the stationary state. Consider a large time $t$. The important term to consider in (46) is the second term. We introduce variables $w = u/t$ and $y = |x - x_r|/t$ so that the second term may be written as

$$r \int_0^t du \, e^{-ru} P_0(x, u | x_r) = \frac{r t^{1/2}}{\sqrt{4\pi D}} \int_0^1 \frac{dw}{w^{1/2}} \exp\left( -t \left[ rw + \frac{y^2}{4Dw} \right] \right). \tag{55}$$

Then the integral is in a form that may be evaluated by Laplace's method (which is a particular case of the saddle point method). Since $t$ is large, the integrand will be sharply peaked at the maximum of the argument of the exponential, which is given by

$$w^* = \frac{y}{\sqrt{4Dr}} \,. \tag{56}$$

Then, if $w^*$ is within the limits of integration, i.e. $w^* \leq 1$, we obtain

$$r \int_0^t du\, e^{-ru} P_0(x,u|x_r) \sim \exp\left(-t[r/D]^{1/2} y\right), \tag{57}$$

where we have ignored the prefactors to the exponential. But if $w^*$ is outwith the limits of integration, i.e. $w^* > 1$, the integral is dominated by $w = 1$ and the contribution of the integral is of the same order as the first term in (46), $e^{-rt} P_0(x,t|x_0)$. See Appendix D for more details).

The condition $w^* < 1$ corresponds to $y = |x-x_r|/t < \sqrt{4Dr}$. Thus there is an equilibration front travelling with speed $v = \sqrt{4Dr}$. When $|x - x_r| < vt$, $P_r(x,t|x_0)$ has relaxed to its stationary form, but if $|x - x_r| > vt$, $P_r(x,t|x_0)$ is still time dependent and is dominated by trajectories in which no resets have occurred or the last reset was near $t' = 0$. See [9] for a more detailed discussion.

We also note that, ignoring prefactors, may write for large $t$ and $|x-x_r|$ with $y = |x-x_r|/t$

$$P_r(x,t|x_0) \sim \exp\left[-tI\left(|x - x_r|/t\right)\right], \tag{58}$$

where

$$I(y) = \begin{cases} (r/D)^{1/2} y, & \text{when } y < \sqrt{4Dr}, \\ r + y^2/4D, & \text{when } y \geq \sqrt{4Dr}. \end{cases} \tag{59}$$

Here $I(y)$ is our first example of a large deviation function (see [9] for more details).

## 5 Diffusion with resetting and absorbing target

We now consider our diffusing particle under resetting as a searcher searching for the target. As in Section 3, we model the target with an absorbing site at $x_T = 0$ [1, 2, 11]. Using a last renewal approach (See Fig. 3) we can write the survival probability for the system with resetting in terms of the system without resetting as

$$q_r(t|x_0) = \underbrace{e^{-rt} q_0(t|x_0)}_{\substack{\text{Probability of surviving} \\ \text{with no reset up to time } t}} + \int_0^t \underbrace{dt' r e^{-r(t-t')}}_{\substack{\text{Probability of reset between} \\ t' \text{ and } t'+dt' \text{ and no} \\ \text{resets for the remaining time.}}} \underbrace{q_r(t'|x_0)}_{\substack{\text{Probability of survival} \\ \text{till } t' \text{ with resetting.}}} \underbrace{q_0(t-t'|x_r).}_{\substack{\text{Probability of survival} \\ \text{without resetting for} \\ \text{the remaining time.}}} \tag{60}$$

Again there are two terms: the first is the probability of no resets and survival, the second integrates over the time $t'$ of the last reset. Recognizing that in (60), the term inside the integral is a convolution, we can make use of the convolution theorem to obtain the Laplace transform of $q_r(t|x_0)$ as,

$$\widetilde{q}_r(s|x_0) = \widetilde{q}_0(s+r|x_0) + r\widetilde{q}_r(s|x_0)\widetilde{q}_0(s+r|x_r). \tag{61}$$

Then we obtain

$$\widetilde{q}_r(s|x_0) = \frac{\widetilde{q}_0(s+r|x_0)}{1 - r\widetilde{q}_0(s+r|x_r)}, \tag{62}$$

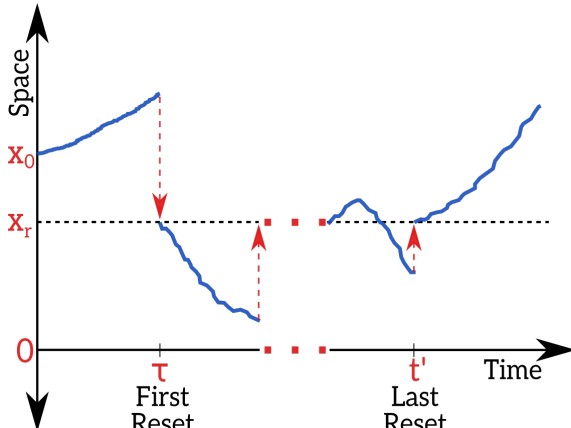

Figure 3: A schematic of a trajectory that does not cross $x = 0$ and hence survives till time $t$. $t'$ represents the time of last reset and $\tau$ represents the time of first reset.

which upon taking the initial and reset position to be the same *ie*, $x_r = x_0$, becomes

$$\widetilde{q}_r(s|x_0) = \frac{\widetilde{q}_0(s+r|x_0)}{1 - r\widetilde{q}_0(s+r|x_0)} \tag{63}$$

$$= \frac{1 - e^{-\alpha(s)x_0}}{s + re^{-\alpha(s)x_0}}, \tag{64}$$

where $\alpha(s) = \sqrt{\frac{r+s}{D}}$, and we have used (16) to obtain (64).

**Note:** Instead of a last renewal equation for the survival probability as given in (60), one could also write an equivalent first renewal equation with $\tau$ representing the time of first reset as (See Fig. 3),

$$q_r(t|x_0) = \underbrace{e^{-rt}q_0(t|x_0)}_{\substack{\text{Probability of surviving} \\ \text{with no reset up to time } t}} + \int_0^t \underbrace{d\tau\, re^{-r\tau}}_{\substack{\text{Probability of a single} \\ \text{reset between } \tau \\ \text{and } \tau + d\tau.}} \underbrace{q_0(\tau|x_0)}_{\substack{\text{Probability of survival} \\ \text{till } \tau \text{ without resetting.}}} \underbrace{q_r(t-\tau|x_r)}_{\substack{\text{Probability of survival} \\ \text{with resetting for} \\ \text{the remaining time.}}} \tag{65}$$

Taking the Laplace transform of (65) then yields an alternative equation for the Laplace transforms

$$\widetilde{q}_r(s|x_0) = \widetilde{q}_0(s+r|x_0) + r\widetilde{q}_0(r+s|x_0)\widetilde{q}_r(s|x_r), \tag{66}$$

which reduces to (61) when $x_0 = x_r$.

## 5.1 Mean first passage time

Using (36), the mean first passage time (MFPT), calculated from (64), reduces to the simple expression

$$\langle T_r(x_0) \rangle = \widetilde{q}_r(s = 0|x_0) \tag{67}$$

$$= \frac{e^{\alpha_0 x_0} - 1}{r}. \tag{68}$$

Expression (68) has diverging behaviour for both the small $r$ and large $r$ limits,

- For $r \ll 1$ $\quad \langle T_r(x_0) \rangle \to \dfrac{x_0}{\sqrt{Dr}}.$

- For $r \gg 1$ $\quad \langle T_r(x_0) \rangle \to \dfrac{e^{\sqrt{\frac{r}{D}}x_0}}{r}.$

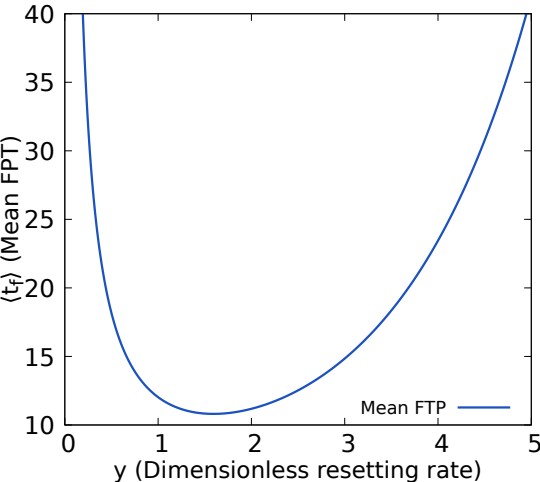

Figure 4: A plot of MFPT for $x_0^2/D = 7$. We observe that the function is a non-monotonic function of $y$ and has a minimum at $y^* = 1.5936\ldots$

The limit $r \to 0$ recovers the diverging diffusive MFPT discussed in Section 3. The limit $r \to \infty$ implies resetting to $x_r$ with infinite rate. The particle is then unable to diffuse away and find the target, and the MFPT diverges. In between these two limits the MFPT has an optimal value of resetting which minimizes the MFPT. This optimal value, $r^*$, can be obtained by evaluating $\frac{d\langle T_r(x_0)\rangle}{dr} = 0$, which results in

$$\frac{y}{2} = 1 - e^{-y}, \tag{69}$$

where $y = \frac{x_0}{(D/r)^{1/2}}$. The dimensionless parameter $y$ is the ratio of the two length scales in the system: $x_0$, the distance of the resetting site to the target (at the origin) and $(D/r)^{1/2}$, the typical distance diffused between resets. Equation (69) has a unique non-zero solution given by $y^* = 1.5936\ldots$, that minimizes the MFPT for searches with resetting. A plot of the MFPT is illustrated in Fig. 4.

Note that here the optimisation of the MFPT assumes that we know the distance to the target $x_0$. Obviously, this requires significant information. Nevertheless, the calculation demonstrates the principle that the MFPT can be optimised by tuning the resetting rate. Also we see that any (finite) resetting rate performs better than the diffusive process without resetting.

## 5.2 Long time behaviour of survival probability

It is difficult to invert the Laplace transform of the survivial probability for all time $t$. For an integral expression see [12]. However, as in the diffusive case discussed in Section 3, the asymptotic behaviour can be easily obtained.

The asymptotic behaviour of the survival probability (64) can be obtained from the singularities of the Laplace transform [1, 2]. The singularity furthest to the right (with largest real part) in the complex $s$ plane dominates the long term behaviour. Expression (64) has two singularities:

(i) Branch point at $s = -r$.

(ii) Pole at $s_0$, which satisfies

$$s_0 + r \exp\left(-\sqrt{\frac{r + s_0}{D}} x_0\right) = 0. \tag{70}$$

Since $s_0 > -r$, the pole is the furthest singularity to the right and the contribution from the pole determines the leading order behaviour,

$$q(t|x_0) \underset{t \gg 1}{\simeq} A e^{s_0 t}, \tag{71}$$

where the constant $A$ is the residue from the pole, which may be calculated to be

$$A = \frac{1 + s_0/r}{1 + s_0 x_r/(4(r+s_0)D)^{1/2}}. \tag{72}$$

If $|s_0| \ll 1$ then (70) yields,

$$s_0 \simeq -r e^{-y}, \tag{73}$$

where $y = \sqrt{\frac{r}{D}} x_0$, which in turn implies $y \gg 1$. Then we have

$$q(t|x_0) \simeq e^{-r t e^{-y}}, \tag{74}$$

which is of the form of the Gumbel distribution.

**Aside on Gumbel distribution**

Consider a situation where $N$ iid random variables are drawn from a PDF, $f(x)$. We are interested in finding the probability that the maximum of these $N$ random numbers is less than a given number $M$. That is,

$$\mathbb{P}(\max_{1 \le i \le N} X_i \le M) = [\mathbb{P}(X_1 \le M)]^N = \left[ \int_{-\infty}^{M} f(x)dx \right]^N = \exp\left( N \ln\left[ 1 - \int_{M}^{\infty} f(x)dx \right] \right). \tag{75}$$

Now if $f(x)$ has an exponential tail, $f(x) \sim A e^{-ax}$ for $x \gg 1$, we obtain the Gumbel distribution from (75),

$$\mathbb{P}(\max_{1 \le i \le N} X_i \le M) \approx \exp\left[ -\frac{NA}{a} e^{-aM} \right]. \tag{76}$$

In the case of the survival of a diffusive particle under stochastic resetting, we have a similar situation [2, 11]. Between each reset the particle must survive, that is, its maximum excursion to the left must be less than $x_0$. The maximum excursion between resets $i-1$ and $i$, is a random variable that we'll call $X_i$, and due to renewal the $X_i$ are independent random variables. For survival through $N$ resets we must have $X_i < x_0$ for $i = 1, \ldots, N$. That is, the maximum of the $X_i$ must be less than $x_0$ which is the problem described above. We obtain the probability that $X_i < x_0$ by averaging the survival probability for a diffusive particle $q_0(t|x_0)$ over the duration of each excursion. Then, with the expected number of resets $rt$ in the long time limit, we obtain the Gumbel distribution through

$$\begin{aligned} q_r(t|x_0) &\approx \left[ \int_0^\infty dt\, r e^{-rt} q_0(t|x_0) \right]^{rt} \qquad \text{(Using (29))} \\ &= \left[ 1 - e^{-x_0 \sqrt{\frac{r}{D}}} \right]^{rt} \\ &\simeq e^{-r t e^{-y}}. \end{aligned} \tag{77}$$

## 5.3 Mini-summary

So far we have seen how stochastic resetting changes the behaviour of diffusion with an absorbing target. First, the mean first passage time is rendered finite with the introduction of resetting whereas it is infinite without resetting. Second, the survival probability decays as an exponential for large time rather than as a power law, as is the case without resetting. The form of the exponential decay can be understood from extreme value statistics. As an aside, we note that in some circumstances resetting can also generate a non-trivial power decay of the survival probability [13]. We now turn to the subject of this summer school: large deviations.

# 6 Large deviations in stochastic resetting

## 6.1 Additive functional

We are interested in calculating the large deviations of an additive functional for a process with resetting [14–16]. Consider a path with $N$ resets. Let $f(x_t)$ be an additive functional of the trajectory, using which we can define

$$
\begin{aligned}
A_t &= \int_0^t dt\, f(x_t) \\
&= \sum_{i=1}^{N+1} A_{(t_n - t_{n-1})},
\end{aligned}
\tag{78}
$$

where $t_n$ is the time of the $n^{\text{th}}$ reset and we have defined $t_0 = 0$ and $t_{N+1} = t$. For simplicity we will assume $f \geq 0$ so that $A_t \geq 0$ (although more general cases can be considered [14–16]).

We expect a large deviation principle to hold with the form

$$
P_r\left(\frac{A_t}{t} = a, t\right) \sim \exp\left(-t I(a)\right),
\tag{79}
$$

where $I(a)$ is the large deviation function, sometimes referred at as the rate function. Using the first renewal formalism, we can write an equation for $P_r(A_t, t)$ as

$$
P_r(A_t, t) = e^{-rt} P_0(A_t, t) + \int_0^t d\tau\, e^{-r\tau} r\left(\int_0^{A_t} dA_\tau P_0(A_\tau, \tau) P_r(A - A_\tau, t - \tau)\right),
\tag{80}
$$

where we have used the additive property of the functional to write a convolution over $A_\tau$. Define a generating function (which in the case of $A_t \geq 0$ is simply a Laplace transform over $A_t$)

$$
G_r(k, t) = \langle e^{-kA_t} \rangle = \int_0^\infty dA_t\, e^{-kA_t} P_r(A_t, t),
\tag{81}
$$

using which we can rewrite (80), making use of the convolution theorem as

$$
G_r(k, t) = e^{-rt} G_0(k, t) + \int_0^t d\tau\, r e^{-r\tau} G_0(k, \tau) G_r(k, t - \tau).
\tag{82}
$$

Now define the Laplace transform of $G_r(A_t, t)$ as

$$
\widetilde{G}_r(k, s) = \int_0^\infty dt\, e^{-st} G_r(k, t),
\tag{83}
$$

from which we obtain again, making use of the convolution theorem,

$$
\widetilde{G}_r(k, s) = \widetilde{G}_0(k, s + r) + r \widetilde{G}_0(k, s + r) \widetilde{G}_r(k, s),
\tag{84}
$$

which can be rearranged to obtain,

$$
\widetilde{G}_r(k, s) = \frac{\widetilde{G}_0(k, s + r)}{1 - r \widetilde{G}_0(k, s + r)}.
\tag{85}
$$

In (85), we have obtained the Laplace transform of the generating function with resetting in terms of the Laplace transform of the generating function without resetting. Equation (85)

has a pole at $1 - r\widetilde{G}_0(k, s+r) = 0$, which we denote by $s_0(k, r)$. So upon inverting, we obtain the leading behaviour of $G_r(k, t)$ as

$$G_r(k, t) \sim e^{s_0(k,r)t} \,. \tag{86}$$

Then upon a second inversion with respect to $k$, we obtain (ignoring sub-exponential terms),

$$P_r(A_t, t) \sim \int dk \, \exp\left( t\left[ \frac{A_t}{t}k + s_0(k, r) \right] \right), \tag{87}$$

which is dominated by the saddle point,

$$\frac{A_t}{t} + \frac{d}{dk}s_0(k, r) = 0\,. \tag{88}$$

So we obtain the large deviation form for $P_r(A_t, t)$ as

$$P_r(A_t, t) \sim e^{tI(a)}\,, \tag{89}$$

where $I(a)$ is the Legendre-Fenchel transform of $s_0$

$$I(a) = -\sup_k (ka + s_0(k, r))\,, \tag{90}$$

with $a = \frac{A_t}{t}$.

## 7   The cost of stochastic resetting

In the model so far, we have assumed that the resets are instantaneous and cost free, but this assumption is unrealistic as resets must come at a price. To rectify this and make the model more applicable, we introduce a cost to each of the resets [17, 18], which may also be thought of as a time penalty.

We consider here the simple case of a diffusing particle under resetting with cost but no absorbing target [18]. A different ensemble of trajectories is to consider the cost of resetting up to absorption by a target at the origin [17].

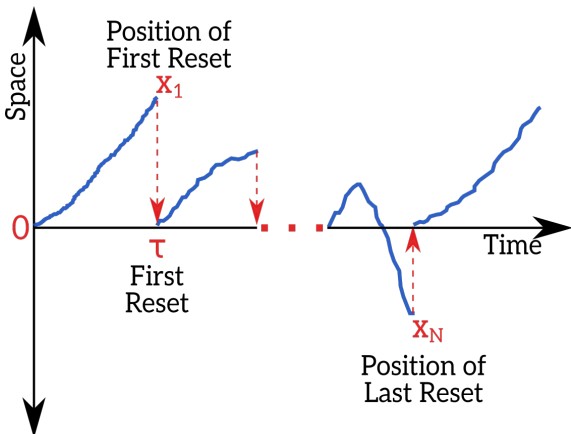

Figure 5: For simplicity, we set the particle to start and reset to $x = 0$. Cost is incurred at the jumps denoted by $x_i$. $\tau$ represents the time of first reset.

## 7.1 Additive cost function

We consider additive costs that occur at the resets (See Fig. 5) of the following form,

$$C = \sum_{i=1}^{N} c(|x_i|), \tag{91}$$

where we take $x_i$ as the position just before the reset and the resetting position as the origin. The resetting costs are functionals of resetting jumps. Example costs include:

- A constant cost $c(x) = c$. This can be thought of as a fixed charge or time penalty such as refractory period for each reset [19].

- Linear cost per reset: $c(x) = \frac{|x|}{V}$. This can be interpreted as the time to return to the origin at a constant speed $V$. [20]

- Quadratic cost per reset $c(x) = x^2$. This is related to "thermodynamics of resetting" [21].

- Exponential cost per reset: $c(x) = e^{|x|}$. This could be used to model situation where resets from a large distance are highly unfavourable [17].

We are interested in calculating the statistics of the total cost, such as the mean $\langle C \rangle$ for the process up to a fixed time $t$ and also the large deviation function.

To begin with the calculation for the mean total cost, we write down a first renewal equation for $\boldsymbol{P}_r(C, t)$, which is the joint PDF of having incurred a total cost $C$ at time $t$. We obtain the first renewal equation as

$$\boldsymbol{P}_r(C, t) = \delta(C)e^{-rt} + \int_0^t d\tau \, re^{-r\tau} \int_{-\infty}^{\infty} dx \, P_0(x, \tau)\boldsymbol{P}_r(C - c(x), t - \tau)\Theta(C - c(|x|)), \tag{92}$$

where $\Theta(C - c(|x|))$ ensures that the total cost is kept positive. The first term accounts for trajectories with no resets and the second term integrates over trajectories where the first reset occurs at time $\tau$.

We then define a generating function

$$G_r(k, t) = \int_0^{\infty} dC \, e^{-kC}\boldsymbol{P}_r(C, t). \tag{93}$$

Using the definition of the generating function on (92), we obtain

$$G_r(k, t) = e^{-rt} + \int_0^t d\tau \, re^{-r\tau} \int_{-\infty}^{\infty} dx \, P_0(x, \tau) \int_0^{\infty} dC \, e^{-kC}\boldsymbol{P}_r(C - c(x), t - \tau)\Theta(C - c(|x|)). \tag{94}$$

Making a change of variable to $C' = C - c(|x|)$, we get

$$\begin{aligned} G_r(k, t) &= e^{-rt} + \int_0^t d\tau \, re^{-r\tau} \int_{-\infty}^{\infty} dx \, P_0(x, \tau) \int_{-c(|x|)}^{\infty} dC' e^{-k(C' + c(|x|))}\boldsymbol{P}_r(C', t - \tau)\Theta(C') \\ &= e^{-rt} + \int_0^t d\tau \, re^{-r\tau} \int_{-\infty}^{\infty} dx \, P_0(x, \tau) \int_0^{\infty} dC' \, e^{-k(C' + c(|x|))}\boldsymbol{P}_r(C', t - \tau) \\ &= e^{-rt} + \int_0^t d\tau \, re^{-r\tau} \int_{-\infty}^{\infty} dx \, P_0(x, \tau)e^{-kc(|x|)}G_r(k, t - \tau). \end{aligned} \tag{95}$$

Finally taking a Laplace transform of (95) by making use of the convolution theorem, we get

$$\widetilde{G}_r(k,s) = \frac{1}{s+r} + r \int_{-\infty}^{\infty} dx \, \widetilde{P}_0(x, r+s) e^{-kc(|x|)} \widetilde{G}_r(k,s), \tag{96}$$

which can be rewritten as

$$\widetilde{G}_r(k,s) = \frac{1}{s+r} \frac{1}{\left[1 - r \int_{-\infty}^{\infty} dx \, \widetilde{P}_0(x, r+s) e^{-kc(|x|)}\right]}. \tag{97}$$

Equation (97) gives an exact expression for the Laplace transform of the generating function of the cost, from which exact expressions for the moments of the cost, can be computed.

## 7.2 Mean total cost

For example, the mean total cost can then be obtained as

$$\langle C \rangle = \mathcal{L}_{s \to t}^{-1} \left[ -\partial_k \widetilde{G}_r(k,s) \Big|_{k \to 0^+} \right] \tag{98}$$

$$= \mathcal{L}_{s \to t}^{-1} \left[ \frac{2r(r+s)}{s^2} \int_0^{\infty} dx \, \widetilde{P}_0(x, r+s) c(|x|) \right]. \tag{99}$$

Performing these calculations for the linear and quadratic cost per reset (see Appendix E), leads to the following mean total costs, which is written in terms of the dimensionless resetting rate $R = rt$.

- Linear cost per resetting: $c(x) = |x|$

$$\langle C \rangle_{\mathrm{lin}} = \sqrt{Dt} \left( \frac{e^{-R}}{\sqrt{\pi}} + \frac{(2R-1)\operatorname{erf}(\sqrt{R})}{2\sqrt{R}} \right). \tag{100}$$

  For large values of $t$ (and therefore $R$), this increases as, $\langle C \rangle_{\mathrm{lin}} \simeq \sqrt{Drt}$.

- Quadratic cost per resetting: $c(x) = |x|^2$

$$\langle C \rangle_{\mathrm{quad}} = \frac{2Dt(R + e^{-R} - 1)}{R}. \tag{101}$$

  For large values of $t$, the mean total quadratic cost increases as $\langle C \rangle_{\mathrm{quad}} \simeq 2Dt$. Interestingly, this does not depend on $r$ and therefore is finite in the limit $r \searrow 0$. The explanation is that in this limit the probability of a reset is $O(rt)$ but the typical cost of a reset is $O(1/r)$, which results in an $O(t)$ contribution to the mean cost as $r \searrow 0$.

The large time behaviours can be explained by the following scaling argument. The distribution of the position $x$ at a reset is given by integrating the Gaussian distribution over the waiting time distribution between resets

$$P(x) \simeq \int_0^{\infty} dt \, \frac{re^{-rt}}{\sqrt{4\pi Dt}} \exp\left(-x^2/4Dt\right) \tag{102}$$

$$= \frac{\alpha_0}{2} e^{-\alpha_0 |x|}, \tag{103}$$

where we have used our trusty formula (16), and as before $\alpha_0 = (r/D)^{1/2}$. Then, for large times, the mean cost will be the mean number of resets $rt$ multiplied by the mean cost per reset. For the case of $c(|x|) = |x|^\beta$ this gives

$$\langle C \rangle = rt \int_0^\infty dx\, \alpha_0 e^{-\alpha_0 x} x^\beta \tag{104}$$

$$= \frac{\Gamma(\beta+1)rt}{\alpha_0^\beta}\,, \tag{105}$$

which precisely recovers the long time limits of the exact results above for the case $\beta = 1, 2$.

## 7.3 Large deviation function

The large deviation function of the cost can be obtained from (97) following the approach of Section 6. Here, we work out the case of linear cost per reset $c = |x|$.

The integral in (97) becomes

$$r \int_{-\infty}^\infty dx\, \widetilde{P}_0(x, r+s) e^{-kc(|x|)} = r \int_{-\infty}^\infty dx\, \frac{1}{\sqrt{4(r+s)D}} e^{-\sqrt{\frac{r+s}{D}}|x|} e^{-k|x|}$$

$$= \frac{r}{r+s+k\sqrt{D(r+s)}}\,, \tag{106}$$

and we obtain

$$\widetilde{G}_r(k,s) = \frac{r+s+kD^{1/2}(r+s)^{1/2}}{(r+s)(s+kD^{1/2}(r+s)^{1/2})}\,. \tag{107}$$

It is easiest to invert the Laplace transform over $k$ first as the only singularity is a simple pole at $k = -s/(D(r+s))^{1/2}$. Therefore we obtain

$$\widetilde{P}_r(C,s) = \frac{r}{(r+s)^{3/2}D^{1/2}} \exp\left(-\frac{sC}{D^{1/2}(r+s)^{1/2}}\right)\,. \tag{108}$$

We now invert the Laplace transform over $s$ by using the Bromwich contour

$$P_r(C,t) = \frac{1}{2\pi i} \int_{-i\infty+\gamma}^{+i\infty+\gamma} ds\, e^{st} \widetilde{P}_r(C,s)\,, \tag{109}$$

where $\gamma$ should be to the right of the singularity at $s = -r$. Setting $C = at$, with $t$ large, we evaluate the inversion integral over $s$ by the saddle point method,

$$P_r\left(\frac{C}{t} = a, t\right) \sim \int_{-i\infty+s_0}^{+i\infty+s_0} ds \exp\left(-t\left[\frac{as}{D^{1/2}(r+s)^{1/2}} - s\right]\right)\,, \tag{110}$$

where we have chosen $\gamma = s_0$ so that the contour goes through the saddle point $s_0$, in a direction parallel to the imaginary axis, and for simplicity we have ignored prefactors to the exponential. The saddle point equation for $s_0$ reduces to

$$r + s_0 = -r + \frac{2D^{1/2}}{a}(r+s_0)^{3/2}\,, \tag{111}$$

which has a unique real solution with $(r+s_0) > 0$.

Therefore $P_r(C/t = a, t) \sim \exp(-tI(a))$ where the large deviation function is

$$I(a) = \frac{as_0}{D^{1/2}(r+s_0)^{1/2}} - s_0\,, \tag{112}$$

where $s_0(a,r)$ is the solution of (111). Note that for $a = (Dr)^{1/2}$ the solution of (111) is $s_0 = 0$ and $I(a = (Dr)^{1/2}) = 0$, which is consistent with the mean cost $\langle C \rangle \simeq (Dr)^{1/2}t$, as we saw in Section 7.2.

# 8 Non-Poissonian resetting

So far we have assumed that the resetting is Poissonian, *i.e.*, we have assumed that the waiting time between resets is distributed exponentially. We now relax this assumption and assume a general waiting time distribution $\psi(t)$ [22–25]. This means that a reset occurs, in time $t \to t + dt$ after the last reset, with probability $\psi(t)dt$. We also define the probability of no resets up to time $t$ as

$$\Psi(t) = \int_t^\infty d\tau \, \psi(\tau). \tag{113}$$

Note that we recover the all the results for Poissonian resetting if we set $\psi(t) = re^{-rt}$. Another situation that is also of interest is the case of deterministic reset (or sharp restart) which is given by $\psi(t) = \delta(t - t_r)$.

Similarly to Section 4.1, in the absence of an absorbing target and with $x_0 = x_r$, we can write the propagator for system with resetting using a first renewal equation as

$$P_r(x,t|x_0) = \Psi(t)P_0(x,t|x_0) + \int_0^t d\tau \, \psi(\tau)P_r(x,t-\tau|x_0). \tag{114}$$

Then using Laplace transform we get,

$$\begin{aligned}
\widetilde{P}_r(x,s|x_0) &= \frac{\int_0^\infty dt \, e^{-st}\Psi(t)P_0(x,t|x_0)}{1 - \int_0^\infty dt \, e^{-st}\psi(t)} \\
&= \frac{\int_0^\infty dt \, e^{-st}\Psi(t)P_0(x,t|x_0)}{s\int_0^\infty dt \, e^{-st}\Psi(t)},
\end{aligned} \tag{115}$$

where the last equality is obtained by performing integration by parts on the denominator. The stationary state will then be given by the coefficient of the $\frac{1}{s}$ term in the small $s$ expansion. So we obtain the stationary state to be

$$P^*(x) = \lim_{s \to 0} \frac{\int_0^\infty dt \, e^{-st}\Psi(t)P_0(x,t|x_0)}{\int_0^\infty dt \, e^{-st}\Psi(t)}. \tag{116}$$

Thus, for the NESS to exist, this limit should not vanish. Therefore, we require

$$\int_0^\infty \Psi(t)dt = \int_0^\infty \tau\psi(\tau)d\tau = \mathbb{E}(\tau) < \infty. \tag{117}$$

That is, for a stationary state to exist for diffusion under resetting we require the mean time between resets to be finite.

## 8.1 Survival probability

Let us also consider the survival probability with an arbitrary waiting time distribution. As we did in Section 5, we can write down a first renewal equation for the survival probability as follows:

$$q_r(t|x_0) = \Psi(t)q_0(t|x_0) + \int_0^t d\tau \, \psi(\tau)q_0(\tau|x_0)q_r(t-\tau|x_r). \tag{118}$$

Then, as usual, taking the Laplace transform and setting $x_0 = x_r$ yields

$$\widetilde{q}_r(s|x_0) = \frac{\int_0^\infty dt \, e^{-st}\Psi(t)q_0(t|x_0)}{1 - \int_0^\infty d\tau \, e^{-st}\psi(\tau)q_0(\tau|x_0)}. \tag{119}$$

Finally, we obtain the MFPT by setting $s = 0$

$$
\begin{aligned}
\langle T_r(x_0) \rangle &= \frac{\int_0^\infty dt\, \psi(t) \int_0^t d\tau\, q_0(\tau|x_0)}{1 - \int_0^\infty d\tau\, \psi(\tau) q_0(\tau|x_0)} \\
&= \frac{\int_0^\infty dt\, \psi(t) \int_0^t d\tau\, q_0(\tau|x_0)}{\int_0^\infty d\tau\, \psi(\tau)(1 - q_0(\tau|x_0))}\,.
\end{aligned}
\tag{120}
$$

## 8.2   Optimising the waiting time distribution

In Section 5.1, for the case of Poissonian resetting, we optimised the MFPT to an absorbing target, with respect to the resetting rate $r$. More generally, we can ask, what is the best choice of waiting time distribution to minimize the mean time for first passage?

It turns out that when the target is at a known distance from the resetting site (in the case of resetting to $x_0$ and a target at the origin this distance is $x_0$) the answer to this question is that a deterministic resetting, $\psi(t) = \delta(t - t_r)$ is the best [23,24,26]. But the resetting period, $t_r$, has to be chosen appropriately and relies on us knowing the distance to the target.

A more realistic problem is when we know where the target ought to be (at the origin say) but the actual distance from this position is a random variable, $x_T$, drawn from a target distribution, $\mathcal{P}_T(x_T)$ centred at the origin [27]. Thus we do not know the position of the target exactly. A sensible strategy for searching for the target would be to reset to the origin and select an optimal waiting time distribution for resetting. The relevant quantity to optimise is then the MFPT averaged over the target position, $x_T$,

$$
\overline{\langle T_r(x_0) \rangle} = \int dx_T\, \mathcal{P}_T(x_T) \langle T_r(x_T) \rangle\,,
\tag{121}
$$

where $\langle T_r(x_T) \rangle$ is given by (120) with $x_0$ replaced by $x_T$.

Although this optimisation problem is beyond the scope of these lectures, we mention that the optimal waiting time distribution depends on the target distribution. For example, in the case of an exponential target distribution $\mathcal{P}_T(x_T) = \beta/2 \exp(-\beta|x_T|)$ the optimal waiting time distribution is exponential, i.e. Poissonian resetting with rate $r = \beta^2 D/4$ [28].

# 9   Conclusion

In these lectures we have presented a detailed account of diffusion with stochastic resetting and highlighted some large deviation principles that apply. We have endeavoured to make the lecture notes self-contained as far as possible so that an interested reader can learn the tricks of the trade. The field of stochastic resetting is constantly expanding and evolving and these notes, whilst not aiming to provide comprehensive coverage of all recent developments, will hopefully provide an entry point to this rich and complex topic.

# Acknowledgments

The authors thank the organisers of the Les Houches School on *Theory of Large Deviations and Applications*. M.R.E. thanks Satya Majumdar, Baruch Meerson and Grégory Schehr for helpful feedback on the lectures.

**Funding information** J.C.S. thanks the University of Edinburgh for the award of Edinburgh Doctoral College Scholarship. We thank Richard Blythe for suggesting the proof of (19) in Appendix B. For the purpose of open access, the authors have applied a Creative Commons Attribution (CC BY) licence to any Author Accepted Manuscript version arising from this submission.

# A  Inverting Laplace transform of Gaussian using the Bromwich integral

The task is to evaluate the inverse Laplace transform using the integral obtained from (16) and (18),

$$P(x,t|x_0) = \frac{1}{2\pi i} \int_{\gamma-i\infty}^{\gamma+i\infty} ds \, e^{st} \frac{1}{\sqrt{4sD}} e^{-\sqrt{\frac{s}{D}}|x-x_0|}. \tag{A.1}$$

To evaluate the Bromwich integral (A.1), we integrate along the contour $\Gamma$, shown in Fig. 6, where we have a keyhole construction around the branch point at $s = 0$. From Cauchy's theorem, since there are no singularities inside the contour, we obtain

$$\frac{1}{2\pi i} \oint_\Gamma ds \, e^{st} \frac{1}{\sqrt{4sD}} e^{-\sqrt{\frac{s}{D}}|x-x_0|} = \frac{1}{2\pi i} \left( \int_{C_\gamma} + \int_{C_1} + \int_{C_a} + \int_{C_\epsilon} + \int_{C_b} + \int_{C_2} \right) dz \, e^{zt} \frac{1}{\sqrt{4zD}} e^{-\sqrt{\frac{z}{D}}|x-x_0|}$$
$$= 0. \tag{A.2}$$

Since we eventually take the limits $R \to \infty$ and $\epsilon \to 0$, the contribution from the integral from $C_1, C_2$ and $C_\epsilon$ goes to zero, as a consequence of Jordan's lemma. So we obtain the required integral as

$$\frac{1}{2\pi i} \int_{\gamma-i\infty}^{\gamma+i\infty} ds \, e^{st} \frac{1}{\sqrt{4sD}} e^{-\sqrt{\frac{s}{D}}|x-x_0|} = -\frac{1}{2\pi i} \left( \int_{C_a} + \int_{C_b} \right) dz \, e^{st} \frac{1}{\sqrt{4sD}} e^{-\sqrt{\frac{s}{D}}|x-x_0|}. \tag{A.3}$$

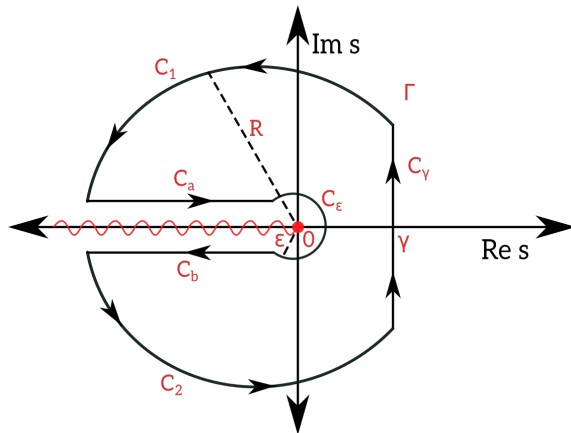

Figure 6: We define the keyhole contour $\Gamma$ composed of $C_\gamma, C_1, C_a, C_\epsilon, C_b$ and $C_2$. $C_\gamma$ is a straight segment that vertically intersects the real axis with all the singularities lying to its left. Here $\gamma > 0$ and $s$ runs from $s = \gamma - iR$ to $s = \gamma + iR$. $C_\epsilon$ is a small circular contour around the branch point singularity at $s = 0$. In the limit $R \to \infty$ and $\epsilon \to 0$, we see that the only contributions to the integral are from $C_\gamma, C_a$ and $C_b$, using which we can calculate the required Bromwich integral.

To evaluate the integral along $C_a$, we choose $s = re^{i\theta}$ with $\theta = \pi$, with the limits of the integral running from $r = \infty$ to $r = 0$, which gives us

$$
\int_{C_a} ds \, e^{st} \frac{1}{\sqrt{4sD}} e^{-\sqrt{\frac{s}{D}}|x-x_0|} = \int_\infty^0 dr \, e^{i\pi} \frac{\exp\left(re^{i\pi}t\right)}{\sqrt{4rDe^{i\pi/2}}} \exp\left(-\sqrt{\frac{r}{D}}e^{i\pi/2}|x-x_0|\right)
$$

$$
= -i \int_0^\infty dr \, \frac{\exp(-rt)}{\sqrt{4rD}} \exp\left(-i\sqrt{\frac{r}{D}}|x-x_0|\right)
$$

$$
= -\frac{2i}{\sqrt{4Dt}} \int_0^\infty du \, e^{-u^2} e^{\frac{iu}{\sqrt{Dt}}|x-x_0|}
$$

$$
= -\frac{2i}{\sqrt{4Dt}} \int_0^\infty du \, e^{-\left(u - \frac{i|x-x_0|}{\sqrt{4Dt}}\right)^2} e^{-\frac{|x-x_0|^2}{4Dt}}
$$

$$
= -i\sqrt{\frac{\pi}{4Dt}} e^{-\frac{|x-x_0|^2}{4Dt}} . \tag{A.4}
$$

To perform the calculations we have used a substitution $u^2 = rt$, followed by completion of the squares and Gaussian integral.

Similarly, to evaluate the integral over $C_b$, we choose $s = re^{i\theta}$ with $\theta = -\pi$, with the limits from $r = 0$ to $r = \infty$. Performing exactly the same steps used to obtain (A.4), we get the same result

$$
\int_{C_b} ds \, e^{st} \frac{1}{\sqrt{4sD}} e^{-\sqrt{\frac{s}{D}}|x-x_0|} = -i\sqrt{\frac{\pi}{4Dt}} e^{-\frac{|x-x_0|^2}{4Dt}} . \tag{A.5}
$$

Plugging in (A.4) and (A.5) into (A.3), we obtain the inverse Laplace transform

$$
P(x,t|x_0) = \frac{1}{2\pi i} \int_{\gamma-i\infty}^{\gamma+i\infty} ds \, e^{st} \frac{1}{\sqrt{4sD}} e^{-\sqrt{\frac{s}{D}}|x-x_0|} = \frac{1}{\sqrt{4\pi Dt}} e^{-\frac{|x-x_0|^2}{4Dt}} . \tag{A.6}
$$

# B Proof of the integral identity (19)

In this appendix we show how to obtain

$$
\int_0^\infty dt \, t^{-1/2} e^{-\frac{\beta}{t} - st} = \frac{\pi^{1/2}}{s^{1/2}} e^{-2(\beta s)^{1/2}} . \tag{B.1}
$$

Note that the above integral converges only when $s > 0$ and $\beta > 0$. We start by defining

$$
I = \int_0^\infty dt \, t^{-1/2} e^{-\frac{\beta}{t} - st} . \tag{B.2}
$$

Using the properties of a Gaussian integral, we can rewrite the integrand in (B.2) as

$$
t^{-1/2} e^{-\frac{\beta}{t}} = \pi^{-1/2} \int_{-\infty}^\infty du \, \exp\left\{\left(-tu^2 + 2i\sqrt{\beta}u\right)\right\} , \tag{B.3}
$$

using which we obtain,

$$
I = \pi^{-1/2} \int_{-\infty}^\infty du \, \exp\left(2i\sqrt{\beta}u\right) \int_0^\infty dt \, \exp\left(-[u^2 + s]t\right)
$$

$$
= \pi^{-1/2} \int_{-\infty}^\infty du \, \frac{\exp\left(2i\sqrt{\beta}u\right)}{u^2 + s} . \tag{B.4}
$$

The last integral can be evaluated using a contour integral along a contour $C$ that runs along the real axis, and is closed in the upper half plane since $\beta > 0$. (This is a consequence of Jordan's lemma.) Thus we obtain,

$$
\begin{aligned}
I &= \pi^{-1/2} \oint_C dz \, \frac{\exp\left(2i\sqrt{\beta}z\right)}{(z - i\sqrt{s})(z + i\sqrt{s})} \\
&= 2\pi^{1/2} i \operatorname{Res}_{z=i\sqrt{s}} \frac{\exp\left(\pm 2i\sqrt{\beta}z\right)}{(z - i\sqrt{s})(z + i\sqrt{s})} \, .
\end{aligned}
\tag{B.5}
$$

Upon evaluating the residue, we obtain

$$
I = \frac{2\sqrt{\pi} i \exp\left(-2\sqrt{\beta s}\right)}{2i\sqrt{s}} = \frac{\pi^{1/2}}{s^{1/2}} e^{-2(\beta s)^{1/2}} \, .
\tag{B.6}
$$

## C   Proof of the Laplace transform of the error function (30)

Using the definition of error function (26), we can re-write (30) as,

$$
\begin{aligned}
\int_0^\infty dt \, e^{-st} \operatorname{erf}\left(\frac{\beta}{t^{1/2}}\right) &= \frac{2}{\sqrt{\pi}} \int_0^\infty dt \, e^{-st} \int_0^{\frac{\beta}{\sqrt{t}}} du \, e^{-u^2} \\
&= \frac{2}{\sqrt{\pi}} \int_0^1 dw \int_0^\infty dt \, t^{-1/2} e^{-st} e^{-\frac{\beta^2 w^2}{t}} \, ,
\end{aligned}
\tag{C.1}
$$

where for the second equality we have performed the substitution, $w = \frac{\sqrt{t}}{\beta} u$ and changed the order of the integration. Now we observe that the second integral in (C.1) is of the form (19) once we make the identification $\beta \to (\beta w)^2$, after which we obtain

$$
\begin{aligned}
\int_0^\infty dt \, e^{-st} \operatorname{erf}\left(\frac{\beta}{t^{1/2}}\right) &= \frac{2\beta}{s^{1/2}} \int_0^1 dw \, e^{-2\beta w s^{1/2}} \\
&= \frac{1 - e^{-2\beta s^{1/2}}}{s} \, .
\end{aligned}
\tag{C.2}
$$

## D   Laplace's method

In this appendix we give the details of evaluating (46) by Laplace's method. Using $w = u/t$ and $y = |x - x_r|/t$, we can rewrite the second term of (46) as

$$
P_r(x, t | x_0) = \frac{e^{-t\phi(1,y)}}{\sqrt{4\pi Dt}} + \frac{r t^{1/2}}{\sqrt{4\pi D}} \int_0^1 \frac{dw}{w^{1/2}} e^{-t\phi(w,y)} \, ,
\tag{D.1}
$$

where

$$
\phi(w, y) = rw + \frac{y^2}{4Dw} \, .
\tag{D.2}
$$

Since we are interested in the large time behaviour, we can evaluate (D.1) using Laplace method, where we expand the integrand around the maxima of the argument of the exponential and integrate. In (D.2) it occurs at the minimum of $\phi(w, y)$, which we obtain as

$$
w^* = \frac{y}{\sqrt{4Dr}} \, .
\tag{D.3}
$$

We get two different large time behaviours for (D.1) depending on whether $w^* < 1$ or $w^* \geq 1$.

For the first case, choose $w^* < 1$, which is the case where the minimum of $\phi(w, y)$ is within the interval $[0, 1]$. Then, the dominant contribution will be from the integral in (D.1). We can evaluate the term by expanding $\phi(w, y)$ to the second order in $w$

$$\phi(w, y) = \phi(w^*, y) + \frac{\phi''(w^*, y)}{2}(w - w^*)^2 + \mathcal{O}(w^3). \tag{D.4}$$

Since $w^*$ is a minimum of $\phi(w, y)$, we have $\phi'(w^*, y) = 0$ and $\phi''(w^*, y) > 0$. Using the substitution

$$s = \sqrt{\frac{t\phi''(w^*, y)}{2}}(w - w^*), \tag{D.5}$$

we obtain

$$\frac{rt^{1/2}}{\sqrt{4\pi D}} \int_0^1 \frac{dw}{w^{1/2}} e^{-t\phi(w,y)} \simeq \frac{re^{-t\phi(w^*,y)}}{\sqrt{2\pi Dw^*\phi''(w^*, y)}} \int_{-\infty}^{\infty} dw\, e^{-s^2}. \tag{D.6}$$

Note that we can safely take the upper and lower limits to $\pm\infty$ as we are interested in the large $t$ behaviour and the contributions from the values different from $w^*$ will be exponentially small. Evaluating all the terms, we obtain

$$P_r(x, t|x_0) \simeq \frac{1}{2}\sqrt{\frac{r}{D}} e^{-\sqrt{\frac{r}{D}}yt}, \tag{D.7}$$

which is the same as the steady-state distribution obtained for diffusion with resetting in (41). Note that the contribution from the first term in (D.1) is sub-leading in $t$ as it will be of the order of $e^{-t\phi(1,y)}/\sqrt{t}$.

In the second case, when $w^* \geq 1$, which occurs when the minimum of $\phi(w, y)$ is outside the interval $(0, 1)$, the major contribution of the integral will be from the boundary close to 1. So the relevant expansion of $\phi(w, y)$ is

$$\phi(w, y) = \phi(1, y) + \phi'(1, y)(w - 1) + \mathcal{O}((w - 1)^2). \tag{D.8}$$

Note that in this case $\phi'(1, y) < 0$. We can approximate the second term in (D.1), by evaluating the integral in the interval $[1 - \epsilon, 1]$. We then obtain

$$\begin{aligned}
\frac{rt^{1/2}}{\sqrt{4\pi D}} \int_0^1 \frac{dw}{w^{1/2}} e^{-t\phi(w,y)} &\simeq -\frac{re^{-\sqrt{4\pi Dt}\phi(1,y)}}{t\phi'(1, y)} \int_0^{-t\phi'(1,y)\epsilon} ds\, e^{-s} \\
&\simeq -\frac{re^{-t\phi(1,y)}}{\sqrt{4\pi Dt}\phi'(1, y)} \int_0^{\infty} ds\, e^{-s} \\
&= -\frac{e^{-t\phi(1,y)}}{\sqrt{4\pi Dt}\phi'(1, y)}. 
\end{aligned} \tag{D.9}$$

To obtain the above expression we have used the substitution $s = t\phi'(1, y)(w - 1)$, and changed the upper limit to $\infty$ as we are interested in the large $t$ behaviour. We see that unlike the previous case, the contribution from the integral and the first term in (D.1) are of the same order. Thus for $w^* \geq 1$, we obtain

$$P_r(x, t|x_0) \simeq \left(\frac{y^2}{y^2 - 4Dr}\right) \frac{e^{-t\left(r + \frac{y^2}{4D}\right)}}{\sqrt{4\pi Dt}}. \tag{D.10}$$

# E Calculation of mean costs for resetting

In the main text we derived

$$\langle C \rangle = \mathcal{L}_{s \to t}^{-1} \left[ \frac{2r(r+s)}{s^2} \int_0^\infty dx \, \widetilde{P}_0(x, r+s) c(|x|) \right], \tag{E.1}$$

where

$$\widetilde{P}_0(x, r+s) = \frac{1}{\sqrt{4D(r+s)}} e^{-((r+s)/D)^{1/2} x}. \tag{E.2}$$

## E.1 Linear cost per reset

For the case of linear cost per resetting, plugging in $c(x) = |x|$ we obtain

$$\int_0^\infty dx \, \widetilde{P}_0(x, r+s) |x| = \frac{D^{1/2}}{2(r+s)^{3/2}}. \tag{E.3}$$

Therefore we obtain from (E.1),

$$\langle C \rangle_{\text{lin}} = r D^{1/2} \mathcal{L}_{s \to t}^{-1} \left[ \frac{1}{s^2 \sqrt{r+s}} \right]. \tag{E.4}$$

Noting that

$$\mathcal{L}_{s \to t}^{-1} \left[ s^{-2} \right] = t, \tag{E.5}$$

and

$$\mathcal{L}_{s \to t}^{-1} \left[ (r+s)^{-1/2} \right] = \frac{e^{-rt}}{\sqrt{\pi t}}, \tag{E.6}$$

we can make use of the convolution theorem to obtain the product of the two Laplace transforms as a convolution, which gives

$$\mathcal{L}_{s \to t}^{-1} \left[ \frac{1}{s^2 \sqrt{r+s}} \right] = \int_0^t d\tau \, (t-\tau) \frac{e^{-r\tau}}{\sqrt{\pi \tau}}$$

$$= \frac{e^{-rt}\sqrt{t}}{\sqrt{\pi} r} + \frac{(2rt-1)\,\text{erf}(\sqrt{rt})}{2r^{3/2}}. \tag{E.7}$$

To obtain the above integral, we have made the substitution $\tau = u^2$ and made use of the definition of the error function (26). Upon plugging (E.7) into (E.4) and defining $R = rt$, we obtain

$$\langle C \rangle_{\text{lin}} = \sqrt{Dt} \left( \frac{e^{-R}}{\sqrt{\pi}} + \frac{(2R-1)\,\text{erf}(\sqrt{R})}{2\sqrt{R}} \right). \tag{E.8}$$

## E.2 Quadratic cost per reset

For the case of quadratic cost per reset, $c(x) = |x|^2$, following exactly the same procedure, we obtain

$$\int_0^\infty dx \, \widetilde{P}_0(x, r+s) |x|^2 = \frac{D}{(r+s)^2}. \tag{E.9}$$

Therefore we obtain from (E.1),

$$\langle C \rangle_{\text{quad}} = 2Dr \mathcal{L}_{s \to t}^{-1} \left[ \frac{1}{s^2(r+s)} \right]. \tag{E.10}$$

Splitting (E.10) into partial fractions, and inverting term by term and again defining $R = rt$, we obtain,

$$\langle C \rangle_{\text{quad}} = 2Dr\mathcal{L}_{s\to t}^{-1}\left[\frac{1}{rs^2} + \frac{1}{r^2(r+s)} - \frac{1}{r^2 s}\right]$$
$$= \frac{2Dt(R + e^{-R} - 1)}{R}. \tag{E.11}$$

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
