# Peer review of "Stochastic Resetting and Large Deviations"

_SciPost Physics Lecture Notes, doi:SciPost Phys. Lect. Notes 103 (2025)_

## Round 1 · Referee Report · Anonymous (Referee 1) · 2025-2-13

Report
The present manuscript represents an interesting lecture notes on stochastic resetting and the large deviation in stochastic resetting, which is of interest to analyze the transition to the nonequilibrium stationary state of the system in the long time limit. It also deals with the cost of resetting. The topic of stochastic resetting has been considered as a hot topic in nonequilibrium statistical physics in the last decade and it is still a mechanism which is used to explain different phenomena, for example, in economy. Therefore, the manuscript is appropriate for publication in SciPost Physics Lecture Notes.
Here are some additional comment to the manuscript.
-
It would be nice if Authors mention that the nonequilibrium stationary state (34) is the Laplace distribution. Graphical representation of it would be useful, from where the cusp at the origin could be visible.
-
It could be useful if the Authors give the renewal equation in Laplace domain, fro $x_0=x_r$, which will yield
$$\tilde{P}_r(x,s|x_r)=\frac{s+r}{s} P_0(x,s+r|x_r),$$from where one can easily find the nonequilibrium stationary state reached in the long time limit, which is actually eq. (40),$$P^{\ast}(x)=r\hat{P}_0(x,r|x_r),$$by using the final value theorem of the Laplace transform,$$\lim_{t→\infty}f(t) = \lim_{s→0}s\hat{f}(s).$$ -
Graphical representation of the MFPT vs resetting rate could be useful in order to see that there is an optimal resetting rate which minimizes the MFPT.
-
When talking about general resetting distribution, useful reference could be [Phys. Rev. E 99, 012141 (2019)].
-
Optional, it could be useful if the Authors show the transition of the mean squared displacement from normal diffusion in the short time limit to saturation in the long time limit due to the resetting of the particle.
Recommendation
Publish (surpasses expectations and criteria for this Journal; among top 10%)

---

## Round 1 · Referee Report · Anonymous (Referee 2) · 2025-2-26

Strengths
Weaknesses
Report
Requested changes
I found a few typos/errors in these notes, which the authors should correct.
-
The factor $\sqrt{s}$ in the prefactor of the expression in equation (13) should be in the numerator, i.e., equation (13) should read $\widetilde{P}(x,s|x_0)=\frac{1}{2}\, \sqrt{\frac{s}{D}}\, e^{-\sqrt{\frac{s}{D}}\, |x-x_0| }$
-
In line 143, "using" should be "Using"
-
With the scaling variable $z=x_0/\sqrt{4 D t}$, equation (21) should read $-2z\frac{dg}{dz} = \frac{d^2 g}{dz^2}$. If the authors want to keep equation (21) unchanged, the correct scaling variable should be $z=x_0/\sqrt{D t}$.
-
The sentence after equation (55) ends with a mathematical expression, and the following sentence again starts with a mathematical variable $y$. It might be better to rephrase the first part of the sentence. For example, "The dimensionless parameter $y$ is the ratio..."
-
In line 289, expression (ii), both $s_0$ and $s$ appear. Is there a typo? Otherwise, give details.
-
In equation (77) and line 370, perhaps $x_i$ in the argument of the theta function should be replaced by $x$.
Recommendation
Publish (surpasses expectations and criteria for this Journal; among top 10%)

---

## Round 1 · Referee Report · Anonymous (Referee 2) · 2025-2-26

Strengths
Weaknesses
Report
Requested changes
I found a few typos/errors in these notes, which the authors should correct.
-
The factor $\sqrt{s}$ in the prefactor of the expression in equation (13) should be in the numerator, i.e., equation (13) should read $\widetilde{P}(x,s|x_0)=\frac{1}{2}\, \sqrt{\frac{s}{D}}\, e^{-\sqrt{\frac{s}{D}}\, |x-x_0| }$
-
In line 143, "using" should be "Using"
-
With the scaling variable $z=x_0/\sqrt{4 D t}$, equation (21) should read $-2z\frac{dg}{dz} = \frac{d^2 g}{dz^2}$. If the authors want to keep equation (21) unchanged, the correct scaling variable should be $z=x_0/\sqrt{D t}$.
-
The sentence after equation (55) ends with a mathematical expression, and the following sentence again starts with a mathematical variable $y$. It might be better to rephrase the first part of the sentence. For example, "The dimensionless parameter $y$ is the ratio..."
-
In line 289, expression (ii), both $s_0$ and $s$ appear. Is there a typo? Otherwise, give details.
-
In equation (77) and line 370, perhaps $x_i$ in the argument of the theta function should be replaced by $x$.
Recommendation
Publish (surpasses expectations and criteria for this Journal; among top 10%)

---

## Round 2 · Author Response

We thank the reviewers for their reports on our submission: “Stochastic Resetting and Large Deviations”. We have made amendments throughout the manuscript (highlighted in blue in the revised version) in response to the comments and suggestions provided by the referees. These changes have improved the quality of our submission and we believe that it is now suitable for publication in SciPost Lecture Notes.
We have attached the revised submission, with the changes highlighted in blue. The list of changes is as follows.
We have attached the revised submission, with the changes highlighted in blue. The list of changes is as follows.

---

## Round 2 · List of Changes

In Response to Referee 1:
Comment 1
It would be nice if Authors mention that the nonequilibrium stationary state (34) is the Laplace distribution. Graphical representation of it would be useful, from where the cusp at the origin could be visible.
Response to Comment 1
Page 9, we have added a sentence to mention that (41) (In the revised version) is the Laplace distribution. We have also added a new figure (Figure 1 in the revised version) which represents the Laplace distribution with a cusp at the resetting position.
Comment 2
It could be useful if the Authors give the renewal equation in Laplace domain, fro x0=xr, which will yield
from where one can easily find the nonequilibrium stationary state reached in the long-time limit, which is actually eq. (40),
by using the final value theorem of the Laplace transform,
Response to Comment 2
Page 10-11, we have added to section 4.1 the suggested alternative derivation of the steady state distribution using Laplace transform and the final value theorem. (48)-(51) includes the suggested equations.
Comment 3
Graphical representation of the MFPT vs resetting rate could be useful in order to see that there is an optimal resetting rate which minimizes the MFPT.
Response to Comment 3
Page 14, we have added figure 4 which shows the plot of MFPT versus y (Dimensionless resetting rate).
Comment 4
When talking about general resetting distribution, useful reference could be [Phys. Rev. E 99, 012141 (2019)].
Response to Comment 4
Page 21, line 437 we have cited the suggested reference as [25].
Comment 5
Optional, it could be useful if the Authors show the transition of the mean squared displacement from normal diffusion in the short time limit to saturation in the long time limit due to the resetting of the particle.
Response to Comment 5
Page 11, we have added a new section 4.2 which calculates the transition of MSD from normal diffusion to the resetting limit.
In Response to Referees 2 and 3:
In Response to Weaknesses:
Comment 1
Since these are meant to be lecture notes (probably to introduce young researchers to the field), it would have been helpful to provide a few intermediate mathematical steps (throughout the paper).
Response to Comment 1
Page 18, we have provided more detailed steps of the calculation for the generating function for the total cost distribution in (94) and (95). We have also made minor changes throughout the submission to improve the readability of the lecture notes. All the other expressions and results which are used in the main text are derived in the appendices.
In Response to Requested Changes:
Comment 1
The factor √s in the prefactor of the expression in equation (13) should be in the numerator, i.e., equation (13) should read
Response to Comment 1
We believe that Equation (13) is correct as stated, since the normalization condition in Laplace space is 1/s. Additionally, we have derived the inverse Laplace transform of Equation (13) and confirmed that it corresponds to the diffusion propagator, as shown in Appendix A.
Comment 2
In line 143, "using" should be "Using"
Response to Comment 2
Page 6, line 146 (In revised version), we have reworded the sentence for better readbility
Comment 3
With the scaling variable z=x0/√4Dt, equation (21) should read-2z dg/dz=d^2 g/dz^2 If the authors want to keep equation (21) unchanged, the correct scaling variable should be z=x0/√Dt.
Response to Comment 3
Page 6, we have corrected equation (27) (In the revised version) to -2z dg/dz=d^2 g/dz^2 , and retained the scaling variable as z=x0/√4Dt.
Comment 4
The sentence after equation (55) ends with a mathematical expression, and the following sentence again starts with a mathematical variable y. It might be better to rephrase the first part of the sentence. For example, "The dimensionless parameter y is the ratio..."
Response to Comment 4
Page 14, we have rephrased the sentence after equation (69) (In the revised version) to read “the dimensionless parameter y is the ratio... ”.
Comment 5
In line 289, expression (ii), both s0 and s appear. Is there a typo? Otherwise, give details.
Response to Comment 5
Page 14, line 310 (In the revised version) we have corrected the typo in (ii) and replaced s with s0.
Comment 6
In equation (77) and line 370, perhaps xi in the argument of the theta function should be replaced by x.
Response to Comment 6
Page 18, line 392 (In the revised version), we have changed the argument inside the theta function to x.
Other changes:
We have made a few minor changes to structuring of the sentences and corrected a few typos throughout the text to improve the readability.
Comment 1
It would be nice if Authors mention that the nonequilibrium stationary state (34) is the Laplace distribution. Graphical representation of it would be useful, from where the cusp at the origin could be visible.
Response to Comment 1
Page 9, we have added a sentence to mention that (41) (In the revised version) is the Laplace distribution. We have also added a new figure (Figure 1 in the revised version) which represents the Laplace distribution with a cusp at the resetting position.
Comment 2
It could be useful if the Authors give the renewal equation in Laplace domain, fro x0=xr, which will yield
from where one can easily find the nonequilibrium stationary state reached in the long-time limit, which is actually eq. (40),
by using the final value theorem of the Laplace transform,
Response to Comment 2
Page 10-11, we have added to section 4.1 the suggested alternative derivation of the steady state distribution using Laplace transform and the final value theorem. (48)-(51) includes the suggested equations.
Comment 3
Graphical representation of the MFPT vs resetting rate could be useful in order to see that there is an optimal resetting rate which minimizes the MFPT.
Response to Comment 3
Page 14, we have added figure 4 which shows the plot of MFPT versus y (Dimensionless resetting rate).
Comment 4
When talking about general resetting distribution, useful reference could be [Phys. Rev. E 99, 012141 (2019)].
Response to Comment 4
Page 21, line 437 we have cited the suggested reference as [25].
Comment 5
Optional, it could be useful if the Authors show the transition of the mean squared displacement from normal diffusion in the short time limit to saturation in the long time limit due to the resetting of the particle.
Response to Comment 5
Page 11, we have added a new section 4.2 which calculates the transition of MSD from normal diffusion to the resetting limit.
In Response to Referees 2 and 3:
In Response to Weaknesses:
Comment 1
Since these are meant to be lecture notes (probably to introduce young researchers to the field), it would have been helpful to provide a few intermediate mathematical steps (throughout the paper).
Response to Comment 1
Page 18, we have provided more detailed steps of the calculation for the generating function for the total cost distribution in (94) and (95). We have also made minor changes throughout the submission to improve the readability of the lecture notes. All the other expressions and results which are used in the main text are derived in the appendices.
In Response to Requested Changes:
Comment 1
The factor √s in the prefactor of the expression in equation (13) should be in the numerator, i.e., equation (13) should read
Response to Comment 1
We believe that Equation (13) is correct as stated, since the normalization condition in Laplace space is 1/s. Additionally, we have derived the inverse Laplace transform of Equation (13) and confirmed that it corresponds to the diffusion propagator, as shown in Appendix A.
Comment 2
In line 143, "using" should be "Using"
Response to Comment 2
Page 6, line 146 (In revised version), we have reworded the sentence for better readbility
Comment 3
With the scaling variable z=x0/√4Dt, equation (21) should read-2z dg/dz=d^2 g/dz^2 If the authors want to keep equation (21) unchanged, the correct scaling variable should be z=x0/√Dt.
Response to Comment 3
Page 6, we have corrected equation (27) (In the revised version) to -2z dg/dz=d^2 g/dz^2 , and retained the scaling variable as z=x0/√4Dt.
Comment 4
The sentence after equation (55) ends with a mathematical expression, and the following sentence again starts with a mathematical variable y. It might be better to rephrase the first part of the sentence. For example, "The dimensionless parameter y is the ratio..."
Response to Comment 4
Page 14, we have rephrased the sentence after equation (69) (In the revised version) to read “the dimensionless parameter y is the ratio... ”.
Comment 5
In line 289, expression (ii), both s0 and s appear. Is there a typo? Otherwise, give details.
Response to Comment 5
Page 14, line 310 (In the revised version) we have corrected the typo in (ii) and replaced s with s0.
Comment 6
In equation (77) and line 370, perhaps xi in the argument of the theta function should be replaced by x.
Response to Comment 6
Page 18, line 392 (In the revised version), we have changed the argument inside the theta function to x.
Other changes:
We have made a few minor changes to structuring of the sentences and corrected a few typos throughout the text to improve the readability.

---

## Editorial Decision

published